# Effect of patient-delivered household contact tracing and prevention for tuberculosis: A household cluster-randomised trial in Malawi

**Kruger Kaswaswa**[1,2☯]*, **Peter MacPherson**[3,4☯], **Moses Kumwenda**[3], **James Mpunga**[2], **Deus Thindwa**[3], **Marriott Nliwasa**[1,3,5], **Mphatso Mwapasa**[1], **Jon Odland**[6], **Tamiwe Tomoka**[1], **Geoffrey Chipungu**[1], **Mavuto Mukaka**[4,7‡], **Elizabeth L. Corbett**[1,3,8‡]

**1** College of Medicine, Blantyre, Malawi, **2** Malawi National TB Control Programme, Blantyre, Malawi, **3** Malawi-Liverpool-Wellcome Trust Clinical Research Programme, Blantyre, Malawi, **4** Department of Clinical Sciences, Liverpool School of Tropical Medicine, Liverpool, United Kingdom, **5** University of Tromso, Tromso, Norway, **6** Centre for Tropical Medicine & Global Health, Nuffield Department of Clinical Medicine, University of Oxford, Oxford, United Kingdom, **7** Faculty of Tropical Medicine, Mahidol-Oxford Tropical Medicine Research Unit, Mahidol University, Bangkok, Thailand, **8** London School of Hygiene and Tropical Medicine, London, United Kingdom

☯ These authors contributed equally to this work.
‡ These authors also contributed equally to the work.
\* kkaswaswa@gmail.com

**Data Availability Statement:** The data that support the findings of this study are openly available as supplement 1.

## Abstract

### Background

Household contact tracing provides TB screening and TB preventive therapy (TPT) to contacts at high risk of TB disease. However, it is resource intensive, inconvenient, and often poorly implemented. We investigated a novel model aiming to improve uptake.

### Methods

Between May and December 2014, we randomised patient with TB who consented to participate in the trial to either standard of care (SOC) or intervention (PACTS) arms. Participants randomised to PACTS received one screening/triage tool (adapted from WHO integrated management of adolescent and adult illnesses [IMAI] guidelines) and sputum pots for each reported household contact. The tool guided participants through symptom screening; TPT (6-months of isoniazid) eligibility; and sputum collection for contacts. Patients randomised to SOC were managed in accordance with national guidelines, that is, they received verbal instruction on who to bring to clinics for investigation using national guidelines.

### Main outcome and measures

The primary outcome was the proportion of adult contacts receiving treatment for TB within 3 months of randomisation. Secondary outcomes were the proportions of child contacts under age 5 years (U5Y) who were commenced on, and completed, TPT. Data were analyzed by logistic regression with random effects to adjust for household clustering.

**Funding:** EL and GC were initials authors who receive the award. The grant number is 2010/859. The funder was Helse Nord TB initiative (HNTI).The funder website is www.helse-nord.no but can also be accessed on http://hnti.medcol.mw/.The funders had no role in study design, data collection and analysis, decision to publish, or preparation of the manuscript.

**Competing interests:** The authors have declared that no competing interests exit.

## Results

Two hundred and fourteen index TB participants were block-randomized from two sites (107 PACTS, reporting 418 contacts; and 107 SOC, reporting 420 contacts). Overall, 62.8% of index TB participants were HIV-positive and 52.1% were TB culture-positive. 250 otherwise eligible TB patients declined participation and 6 households (10 PACTS, 6 SOC) were lost to follow-up and were not included in the analysis. By three months, nine contacts (PACTS: 6, [1.4%]; SOC: 3, [0.7%]) had TB diagnosed, with no difference between groups (adjusted odds ratio [aOR]: 2.18, 95% CI: 0.60–7.95). Eligible PACTS contacts (37/96, 38.5%) were more likely to initiate TPT by 3-months compared to SOC contacts (27/101, 26.7%; aOR 2.27, 95% CI: 1.04–4.98). U5Y children in the PACTS arm (47/81 58.0%) were more likely to have initiated TPT before the 3-month visit compared to SOC children (36/89, 41.4%; aOR: 2.31, 95% CI: 1.05–5.06).

## Conclusions and relevance

A household-centred patient-delivered symptom screen and IPT eligibility assessment significantly increased timely TPT uptake among U5Y children, but did not significantly increase TB diagnosis. This model needs to be optimized for acceptability, given low participation, and investigated in other low resource settings.

## Clinical trial registration

TRIAL REGISTRATION NUMBER: ISRCTN81659509 https://www.isrctn.com/ISRCTN81659509?q=&filters=conditionCategory:Respiratory,recruitmentCountry:Malawi,ageRange:Mixed&sort=&offset=1&totalResults=1&page=1&pageSize=10&searchType=basic-search. 19 July 2012.

## Introduction

An estimated 10 million people developed tuberculosis (TB) and 1.4 million died from the disease in 2019 [1]. TB is the most common presenting illness among people living with HIV, including those taking antiretroviral therapy (ART) and is the major cause of HIV-related deaths, globally [1–4]. Household contacts of adults with pulmonary TB are at risk of TB infection and subsequent active disease. Young children and HIV-positive people exposed through household contact are extremely vulnerable to TB infection and primary progressive TB disease, particularly when they have untreated HIV infection. The prevalence of active TB among household contacts is estimated to be 3.1% [5,6]. In rural South Africa, prevalence of TB in household contacts living with a known TB case was high compared to household contacts living in households without a known TB case (6,075 per 100,000 versus 407 per 100,000), with most contacts with culture-confirmed TB being asymptomatic [7–9]. The pooled yield of TB diagnosis in among children undergoing contact tracing investigations varies, with children younger than five years (8.5%) years being at higher risk than older age groups [10,11].

Systematic screening for TB–including household contact tracing–is likely to be more effective than passive case-finding alone both for individuals and for improving TB epidemiology [9,12–14]. However, household contact tracing is challenging to implement, especially in low-resource settings [15–18]. In Malawi, similar to other countries in the sub-Saharan African

region, household contact tracing is recommended in national policy, but has been suboptimally implemented due to resource limitations (19). In routine practice in many settings, health workers advise index TB patients to bring their contacts to the health facility for symptom screening, followed by clinical investigations (sputum smear microscopy, chest radiography), if symptomatic [17,19,20]. Low update of facility-based household contact screening is likely to be explained by the high transport and opportunity costs associated with facility-based services and completing the screening process [19].

Interventions to increase uptake of TB screening and improve access to TB preventive therapy (TPT) among household contacts are urgently needed. Conventional models of facility-based or outreach services are highly acceptable but have limited capacity to scale-up to optimal levels. Alternatives models of health service delivery that make more use of households participation need to be explored to complement the shortfall of the conventional approaches. Alternative models such as "patient-delivered" strategies have potential to relieve patients of the financial costs that are associated with contact investigations done at facilities.

We therefore undertook a household cluster-randomised trial to investigate the effectiveness of a low-cost, patient-delivered household screening intervention, with the objective of detecting undiagnosed TB among close household contacts of TB patients and improving access to TPT. The main objective was to identify a potentially sustainable model that could be used as an alternative to facility-based contact tracing and screening to improve outcomes for household TB contacts.

## Methods

### Study design

An open label, parallel group cluster-randomised trial.

### Study site and participants

Blantyre District is a major urban centre located in the Southern Province of Malawi and was home to an estimated 956,898 people in 2017 [21]. Adult HIV prevalence was 18.5%, and TB prevalence was estimated at 988 per 100,000 [22].

In Malawi, TB diagnosis, registration and treatment is provided by the National Tuberculosis Programme (NTP), with services available without charge at most hospitals and health centres. We selected as study clinics two health facilities in Blantyre (Queen Elizabeth Central Hospital, and Ndirande Health Centre) that recorded the greatest number of TB case notifications in the city in 2013 [NO_PRINTED_FORM].

Adults with pulmonary TB who registered for treatment at either site between 1st May 2014, and 30th December 2014 were screened for eligibility to be included in the trial. Inclusion criteria were: never previously having been treated for TB; aged 18 years or older on the day of TB registration; at least one child aged five years or younger residing in their households (this criteria removed in a subsequent protocol amendment); resident within Blantyre; and, if an in-patient, were likely to be discharged within two weeks. We excluded participants who were unwilling to provide informed consent, hospitalised patients unlikely to be discharged within 14 days (e.g. due to retreatment, transferred in and out of recruiting facility), and membership of a household already recruited into the study.

We defined a household to be a group of people who had lived in the same dwelling as the index case and who had shared meals or slept in the same room as the index case on most days of the preceding week. Participants completed a clinical and sociodemographic questionnaire and provided details of their current household members to Research Assistants.

## Interventions: Standard of care

Participants allocated to the SOC group were provided with oral advice from study research assistants that they should encourage all household members to report to the health facility for TB screening and HIV testing in accordance with national guidelines. Participants were additionally advised to return to the facility after 28 days with their children. HIV testing and linkage to ART was offered to all index cases through the routine system.

## Interventions: Patient-delivered household active case finding for TB (PACTS)

In addition to interventions received by the SOC group, in the intervention group (patient-delivered household active case finding for TB, PACTS) research assistants provided participants with verbal and written (in Chichewa) information and instructions on how to conduct a symptom screen for TB for all members of their households. Participants were given one TB screening checklist and "contact referral card" per household member, and asked that, upon their return to home, they conducted a TB symptom screen for each household member, recording demographic characteristics and the presence of cough of any duration, fever, weight loss, or night sweats. Participants were additionally provided with a supply of sputum cups, each with a prefilled laboratory request form and specimen bag. Using pictorial tools and verbal and written instructions, Research Assistants instructed participants to support adult (18 years or older) household members with any one or more of TB symptoms to produce a single sputum sample, with specific instruction about sputum collection in well-ventilated outdoor spaces.

Participants were asked to return to the study clinics at their subsequent planned routine TB treatment appointment (usually within 28 days), along with completed household contact screening cards and sputum samples from each symptomatic household member. We additionally asked that the participant bring all children under five years of age to this appointment for clinical assessment.

## Clinic interventions: Both groups

At the study clinic, the trial physician identified all household contacts who returned to the clinic using study self-referral slips. Household contacts were assessed for the presence of TB symptoms by a study clinician without reference to allocation group and were offered HIV testing and TB testing including sputum smear microscopy, sputum Xpert MTB/Rif, and chest x-ray, if indicated, in accordance with Malawi national guidelines [20]. Sputum samples collected from household contacts at clinic assessment were transported to the TB Research Laboratory at the College of Medicine, University of Malawi, Blantyre for fluorescence microscopy and mycobacterial growth inhibitor tube (MGIT) culture, with laboratory staff blinded to allocation group. Household contacts with microbiologically-confirmed TB, or where the health worker made a clinical diagnosis of TB, were supported by research assistants to register for TB treatment at the same facility, with home tracing if required.

After assessment for contraindications, children with a negative TB symptom screen (and negative TB investigations, if done) were initiated onto 6-months of TPT (isoniazid preventive therapy, IPT), with dosing guided by the Malawi National TB Programme, with adherence counselling and monthly (or earlier if required) clinic appointments for follow-up through the routine National TB Programme.

## Outcome assessment

All index case participants were contacted by telephone three months after recruitment, and an appointment was made for a household visit, with tracing supported using a previously

validated electronic geolocation system [23]. During the household visit, all household members (including the index case) were interviewed and underwent a TB symptom screen if not taking TB treatment. Sputum samples for microscopy and MGIT culture were collected from adult household members with TB symptoms (any of cough, fever, weight loss, weight sweats), and both adults and children with TB symptoms were referred to the study clinic for assessment for TB treatment, including clinical assessment, sputum investigation and chest x-ray if required.

We additionally determined whether children aged five years or younger were taking IPT by inspecting medication bottles and patient carried treatment cards during the 3-month assessment. At the study clinics, household members with microbiologically confirmed or clinically-diagnosed TB were supported to access treatment, and children under 5 years in whom TB was excluded were initiated onto IPT. For all household contacts initiated onto IPT, we extracted data from the facility IPT registers for the subsequent 6-month treatment period.

## Definitions and outcomes

We defined two primary trial outcomes. The first primary outcome was the proportion of household contacts with either microbiologically-confirmed or clinically-diagnosed TB at three months after recruitment of the index case participant. This analysis was limited to contacts not already taking TB treatment at the time of TB treatment registration by the index case. Microbiologically-confirmed TB was defined as: at least one positive sputum smear result, or at least one specimen culture-positive for *M. tuberculosis*. Clinically-diagnosed TB was defined through examination of patient-held medical records, with written clinical evidence that a decision to initiate anti-TB treatment had been made accepted, regardless of whether treatment had been started.

The second primary outcome was the proportion of household contacts aged five years or younger who initiated IPT between recruitment of the index case participant and the three-month outcome visit, excluding children reported to be taking IPT or TB treatment at baseline. IPT initiation was defined by documented evidence in clinical records, the study clinic IPT register, or in patient-carried treatment cards that a clinical decision to start IPT had been made, and that at least one month of IPT had been prescribed.

The secondary outcomes were: (i) the proportion of household contacts not taking TB treatment at baseline with microbiologically-confirmed TB diagnosed between recruitment of the index case and the three month outcome assessment; (ii) the proportion of household contacts not taking TB treatment at baseline who completed a TB symptom screen within three months; (iii) the proportion of household contacts not taking TB treatment at baseline with any TB symptoms at three month assessment; (iv) the proportion of household contacts not taking TB treatment at baseline with microbiologically-confirmed TB at three month assessment; and (v) the proportion of household contacts aged 5 years or younger and who initiated IPT who subsequently completed a six month course of IPT within 9-months, defined as having documented evidence of collecting at least six monthly supplies of IPT in the study clinic IPT register.

## Randomisation and blinding

Index case participants and their households were randomised to either the PACTS or SOC group in a 1:1 ratio using block randomisation, with block size of four. Block randomisation codes were pre-generated by the study statistician using the uniform distribution function with a pre-defined seed set in Stata version 13.0. The block randomisation schedule was provided to a data manager independent of the trial and based at the central research offices. At

each study clinic, once consent to participate was obtained, the recruiting research assistant would obtain a randomisation code and allocation group by telephoning the central data manager. Research assistants collecting trial outcomes were blinded to trial arm when conducting home visits, and investigator blinding was maintained until final analysis. No interim analysis was conducted.

## Statistical methods

Our initial sample size was 424 households to provide 80% power to detect at least a 2.5-times relative difference in TB diagnosis, comparing between groups. This assumed a between cluster coefficient of variation of $k$ = 0.30 [24,25], four household members per cluster, a cumulative incidence of active TB of 2% among household contacts allocated to the SOC group, and 20% loss-to-follow-up. As discussed in results, this sample size was not achieved due to the unanticipatedly high refusal to participate by otherwise eligible index patients.

We reported baseline characteristics of index case participants and their household contacts, stratified by allocated group. Outcomes were analysed by intention-to-treat, with all household members allocated to each group (excluding those reported to be taking TB treatment at time of index case recruitment) included in denominators [26]. For each outcome, we constructed a logistic regression model with a random-effects term to account for clustering between households to estimate odds ratios and 95% confidence intervals. In accordance with the protocol, as there was imbalance between groups in the proportion of households in the poorest quintiles and the education level of the household head, we additionally report multivariable odds ratios adjusted for these variables. We further analysed the second primary outcome using the time to event analysis method to investigate the delay in IPT initiation among children under 5 years old. We used the intervention arm as our reference groups to interpret the odds ratio during the analysis.

## Ethical statement

The study was approved by the College of Medicine Research Ethics Committee (COMREC), University of Malawi (P.05/12/1210). Written (or independently witnessed thumbprint if illiterate) informed consent was obtained from all index TB cases. COMREC approved a waiver of consent for recruitment of household members. Prior to outcome assessment at the three-month household visit, we obtained written (or independently witnessed thumbprint) consent from all adult (18 year or older) household members, additional parental/guardian assessment for adolescents aged between 10 and 18 years, and parental/guardian consent for children aged 0–10 years. COMREC also granted a waiver of written informed consent/assent for children younger than 5 years initiating IPT, as this was "an international SOC" and national policy in Malawi. The authors confirm that all ongoing and related trials for this drug/intervention are registered. We delayed in registering the study before recruitment because collaborators wanted to identify a suitable acronym to be used for registration and finalise other administrative requirements for the study.

## Results

Between May 1 and December 30, 2014, 742 index TB cases were assessed for eligibility, 464 met eligibility criteria, 214 were provided informed consent to participate and were randomly allocated to a trial group. One household in the PACTS group did not receive study interventions and was excluded from analysis (Fig 1). There were 849 household contacts that were reported by index cases at the time of recruitment. However, at the time of the evaluation 11 household contacts could not be traced. Thus, the 213 index TB cases had a total of 838 household contacts, 168 of whom were children aged under 5 years. A total of 418 household

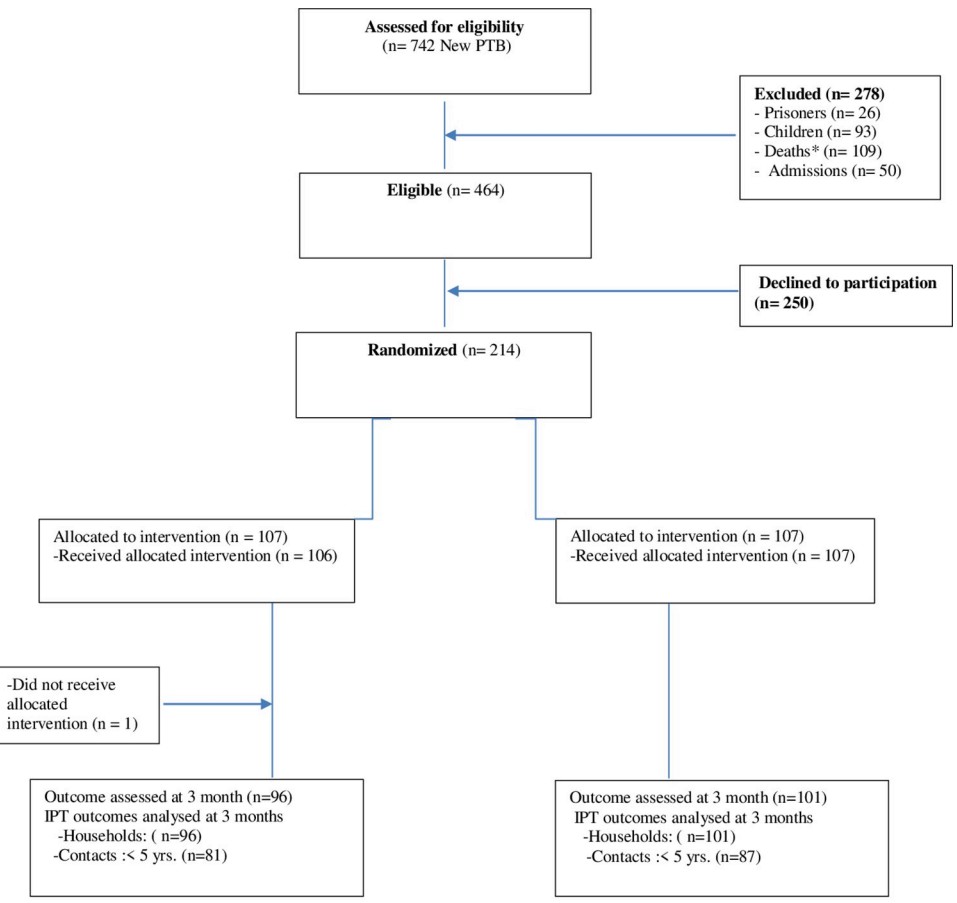

**Fig 1. Trial flow chart.**

contacts in the PACTS group and 420 household contacts in the SOC group were analysed according to protocol in the intention-to-treat analysis.

TB index case participants were predominantly male, and HIV-positivity just under 60% in both arms (Table 1). The median household size was five people, with 66% reporting having at least one child aged younger than five years of age, and a median of two children under five years of age per household. Other index case, household and contact details were balanced between groups, apart from sputum microbiology results (where index cases in the SOC arm were more likely to have microbiologically-confirmed disease than those in the PACTS arm), and household wealth quintile, where a smaller proportion of households in the PACTS group were in the poorest wealth quintile.

In the PACTS group, 96/106 (90.6%) households and 418 household contacts were evaluated at three months. In the SOC group, 101/107 (94.4%) households and 420 household contacts were evaluated. Main reason for contacts not being evaluated was death, with six in the PACTS group and eight in SOC group.

## Primary outcome 1: Tuberculosis diagnosis

Overall, a total of nine household contacts were diagnosed with TB during follow-up: three (0.7%) in the PACTS group and six (1.4%) in the SOC group (Table 2). There was no

**Table 1. Baseline characteristics of participants, household members and household contacts.**

| Characteristic | PACTS: n (%) | SOC: n (%) |
|---|---|---|
| **TB patient characteristics** | | |
| Participants | 106 | 107 |
| Men (%) | 67 (63.2) | 70 (65.4) |
| Age in years, median (range) | 33.5 (19–73) | 32.0 (18–70) |
| Marital Status, No (%) | | |
| Living as if married | 5(4.7) | 5(4.7) |
| Married | 57(53.8) | 61(57.0) |
| Married but not living together | 3 (2.8) | 7 (6.5) |
| Polygamous marriage | 13 (12.3) | 10 (9.3) |
| Separated | 4 (3.8) | 0 (0.0) |
| Single | 7 (6.6) | 5 (4.7) |
| Widowed | 17 (16.0) | 19 (17.8) |
| Education (%) | | |
| No school | 3 (2.8) | 4 (3.7) |
| Primary | 40 (37.7) | 44 (41.1) |
| Secondary | 58 (54.7) | 56 (52.3) |
| Higher | 5 (4.7) | 3 (2.8) |
| Employed | 69 (65.1) | 77 (72.0) |
| HIV positive[a] | 62 (59.6) | 61(58.7) |
| ART treated[b] | 39 (54.2) | 40 (57.1) |
| Sputum bacteriology[*, c] | | |
| S+ve culture +ve | 56 (52.8) | 64 (59.8) |
| S-ve culture +ve | 50 (47.2) | 43(40.2) |
| **Household characteristics** | | |
| Household size, median (range) | 5 (2–19) | 5 (0–11) |
| Children, median (range) | 2 (0–7) | 2 (0–8) |
| Has U5Y old, No. (%) | 66 (62.3%) | 66 (61.7%) |
| Age of household head in years, Median (range) | 36.5 (21.0–80) | 37.0 (18–77) |
| Household wealth quintiles[**, d], No. (%) | | |
| Poorest | 15 (15.8) | 24 (24.0) |
| Poor than average | 16 (16.8) | 23 (23.0) |
| Average poor | 19 (20.0) | 20 (20.0) |
| Wealthier than average | 22 (23.2) | 17 (17.0) |
| Least Poor | 23 (24.2) | 16 (16.0) |
| Previous TB history, No. (%)[f] | 12 (12.5) | 13 (12.9) |
| **Household contacts characteristics** | | |
| Contacts | 418 | 420 |
| Male[g] | 197 (47.2) | 190 (45.2) |
| Female | 220(52.8) | 230 (54.8) |
| Age contact, median (range) | 16.8(1–87) | 13.0 (0–84) |
| Age-groups: contacts (years) | | |
| 0–5 | 81 (19.4) | 87 (20.2) |
| Contact level of education[e] | | |
| No school | 256 (61.8) | 284 (67.6) |
| Primary | 75 (18.1) | 76 (18.1) |
| Secondary | 68 (16.4) | 50 (11.9) |

(*Continued*)

**Table 1.** (Continued)

| Characteristic | PACTS: n (%) | SOC: n (%) |
|---|---|---|
| Higher | 15 (3.6) | 10 (2.4) |

* Abbreviations: *S+ve culture +ve = smear positive culture positive, S-ve culture +ve = smear negative culture positive, S-ve culture–ve = smear negative culture negative.*

** Wealth score was derived from using the Malawi Proxy means tests from the 1997–98 Malawi Integrated Household Survey.

[a]Missing Values: PACTS:9, SOC:9.

[b]Missing Values: PACTS:34, SOC:37.

[c]Missing Values: PACTS:0, SOC:0.

[d]Missing Values: PACTS:11, SOC:7.

[e]Missing Values: PACTS:4, SOC:0.

[f]Missing Values: PACTS:10, SOC:6.

[g]Missing Values: PACTS:1, SOC:0.

statistically significant difference in TB diagnosis between groups in unadjusted (OR: 2.02, 95% CI: 0.48–8.51) or adjusted comparisons (OR: 2.18, 95% CI: 0.60–7.95). Of these, 3.3% of households (7/214) had TB diagnosed before their home visit (3 PACTS, 4 SOC): unadjusted risk ratio (OR: 0.74, 95% CI: 0.17–3. 28) (Table 3).

### Primary outcome 2: IPT initiation among under five-year-old children

Using the numbers of U5Y children reported to be household contacts by index cases at recruitment, 47/81 (58.0%) in the PACTS had initiated IPT by three-month assessment. This compared to 36/87 children (41.4%) in the SOC group. In both unadjusted (OR: 2.02 95% CI: 0.96–4.24) and adjusted analysis (OR: 1.79, 95% CI: 0.80–4.02) there were no significant differences between groups (Table 2).

### Secondary outcomes

Within the nine months following index case TB diagnosis, a total of 50 of 197 households with ≥1 contact aged ≤5years (25.4%) completed IPT at household level. Between the groups,

**Table 2. Effect of interventions on primary and secondary outcomes, at individual-level.**

| Outcome | PACTS n/N (%) | SOC n/N (%) | Odds ratio | 95% CI | Adjusted odds ratio | 95% CI |
|---|---|---|---|---|---|---|
| *Primary outcomes* | | | | | | |
| Proportion of household contacts diagnosed with TB | 3/418 | 6/420 | 2.02 | 0.48–8.51 | 2.18 | 0.60–7.95 |
| Proportion of under 5-year old household contacts who initiated IPT | 47/81 | 36/87 | 2.02 | 0.96–4.24 | 1.79 | 0.80–4.02 |
| *Secondary outcomes* | | | | | | |
| Proportion of under 5-year old household contacts who completed IPT within 9 months | 39/81 | **30/87** | 1.76 | 0.90–3.44 | 1.31 | 0.61–2.85 |
| Proportion of household contacts with symptoms of TB at 3-month assessment | 46/418 | 66/420 | 0.66 | 0.39–1.12 | 0.66 | 0.39–1.10 |
| Proportion of household contacts with microbiologically-confirmed TB at 3-month assessment | 3/418 0.7 (0.1, 2.1)* | 0/420 0 (0, 0.9)* | | | | |
| Proportion of households completing TB symptom screen by 3-month assessment by either: sputum or chest x-ray tests | 20/96 | 10/101 | 2.39 | 1.06–5.43 | 2.58 | 1.12–5.97 |

*Binomial exact %(95% CI).Adjusted for clustering and wealth status. The reference group for the outcome is PACTS.

**Table 3. Primary and secondary outcomes by trial arm, at household-level.**

| | Events: n/N | | Univariate logistic regression | | Multivariate logistic regression* | |
|---|---|---|---|---|---|---|
| Outcomes | PACTS | SOC | OR** | 95% CI | OR | 95% CI |
| ≥1TB case in household initiated before 3 month (1⁰ outcome) | 3/96 | 4/101 | 0.74 | 0.17–3.28 | - | - |
| ≥1 under 5-year old child in household initiated IPT within 3 months in (1⁰ outcome) | 37/96 | 27/101 | 1.94 | 1.07–3.53 | 2.27 | 1.04–4.98 |
| ≥1 under 5-year old child in household completed IPT within 9 months | 28/96 | 22/101 | 1.63 | 0.79–3.38 | 1.78 | 0.80–3.93 |
| ≥1 Contacts screened with sputum or X-rayed for TB before 3 months | 20/96 | 10/101 | 2.39 | 1.05–5.44 | - | - |

\* = Multivariate: adjusted for difference between arms in wealth score and culture

\*\* The reference group for the outcome is PACTS.

there was no significant difference in the IPT completion between PACTS (28/96, 29.1%) and SOC (22/101, 21.7%; OR: 1.78; 95% CI: 0.80–3.93) arms. At individual level, a greater proportion of contacts completed IPT in the PACTS group (39/81, 48.1%) than the SOC group (30/87, 34.4%), but this difference was not significantly different (aOR, 1.31, 95% CI: 0.61–2.85) (Tables 2 and 3).

In the PACTS group, 11.0% (46/418) of household contacts had TB symptoms at 3 months compared to 15.7% (66/420) in the SOC group (aOR 0.66; 95%CI, 0.39–1.10). In the PACTs group 0.7% (3/418) household contacts had untreated microbiologically-confirmed TB compared to 0.0% (1/420) in the SOC group (aOR 2219.02, 95%CI, 7.10–6933.61).

## Time to event of IPT initiation

In an unadjusted but not pre-set analysis, the mean time to IPT initiation was 1.55 months (95% CI: 36.6–54.9 days) in the PACTS arm while the mean time to initiating IPT was higher in the SOC (1.87 months (95% CI, 45.8–64.15). When adjusted for clustering, IPT initiation significantly favoured the PACTS arm (IRR: 1.67 (95% CI 1.07–2.62) (Fig 2).

## TB screening events and IPT eligibility assessments

After 7 months of providing screening and IPT eligibility assessment, 30 of 197 households (15.2%) had at least one household contact investigated by either sputum tests or chest X-ray. Of these, 20/96 (20.8%) were from PACTS group and 10/101 (9.9%) SOC, with the difference remaining significant after adjustment for clustering (aOR 2.58, 95%CI: 1.12–5.97—Table 2).

## Discussion

In this study, we investigated the effectiveness of a low-cost patient-delivered household screening intervention. Although there was low cumulative burden of TB in both arms, evaluation of the patient-delivered approach showed that screening of household contacts has potential to improve the performance of household contact screening from the perspective of increasing the uptake of TB screening investigations (sputum examination and radiography) and initiation of TPT among children under five years of age, with reduced time to TPT initiation. However, this patient-centred approach did not increase detection of undiagnosed active TB or improve the completion rates of IPT among childhood household contacts.

To our knowledge, this is the first cluster randomised trial to investigate the effects of a patient-delivered screening and TPT assessment. Initiation of IPT during the first three

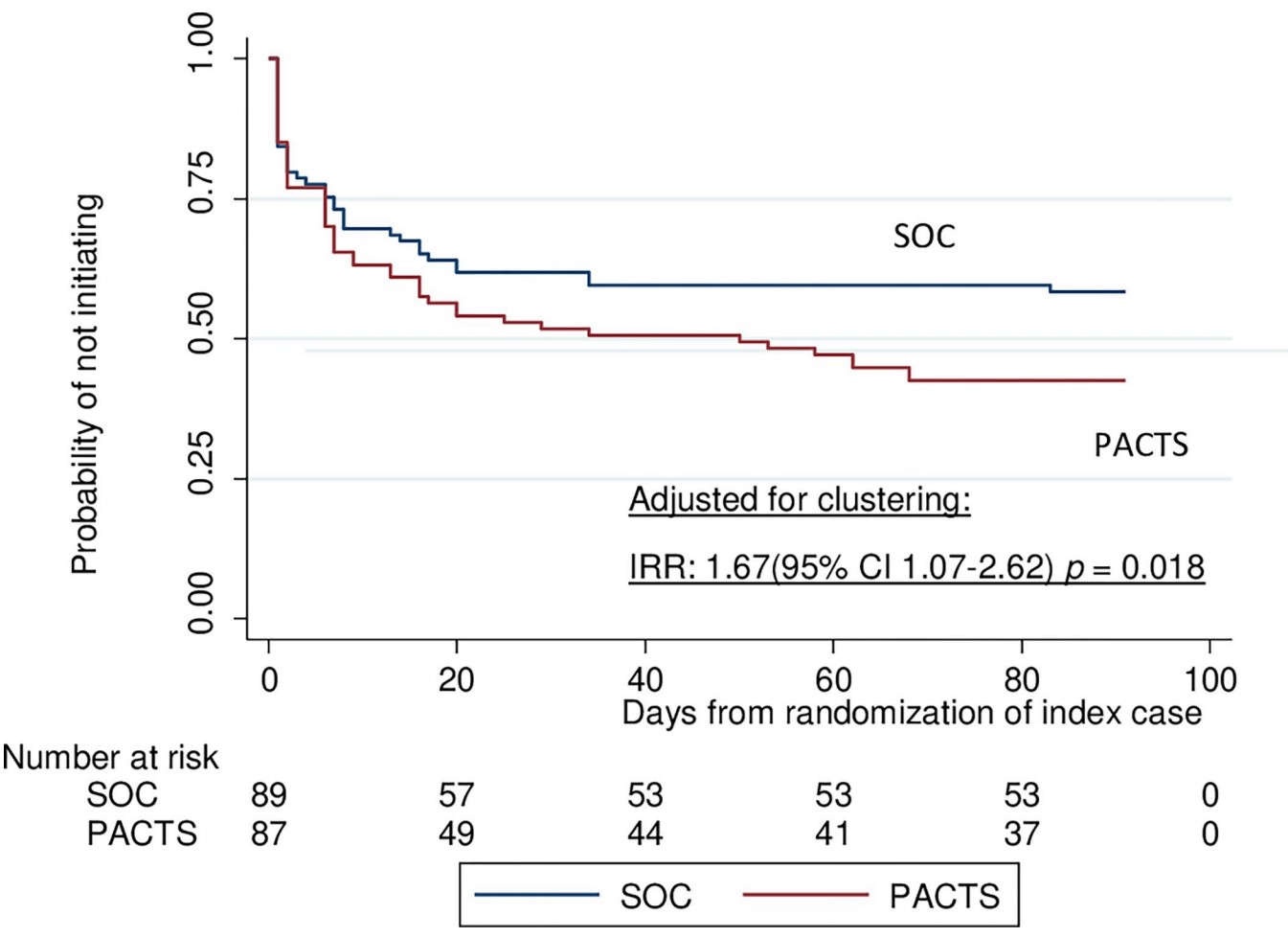

**Fig 2. Probability of not initiating IPT by 3 months after index diagnosis.**

months after diagnosing the index case was higher in the patient-delivered compared to the SOC at household level (38.5% versus 26.7% respectively). At individual level, IPT initiation significantly favoured patient-delivered compared to SOC (58.0% versus 41.4%). At a time when index TB patients may already be facing catastrophic costs due to accessing TB care, and so potentially reluctant to incur any additional costs, screening of contacts with chest X-ray or sputum examination was significantly increased under this patient-delivered approach. This suggests that index patients may have been more willing to invest in the health of their household members when symptomatic screening and IPT eligibility are conducted at home [29].

The yield of bacteriologically confirmed TB was lower in both arms (overall 0.4%, 3/838) than reported from South African studies, but within the expected range for bacteriologically confirmed TB in other African settings [27]. The overall prevalence of undiagnosed bacteriologically confirmed or active TB among the household contacts was 1.07% (9/838) compared to meta-analysis estimates of 1.4% (95% CI: 1.1–2.1) reported for low and middle-income countries [5].

To maximise the uptake of IPT, policy makers should be aware that uptake of IPT may increase further if there is promotion of HIV testing to TB contacts with explicit inclusion of PLHIV as IPT eligible within the Malawian NTP. When an index TB case is also a PLHIV, there is greater likelihood that people living in the same household are HIV-positive, and if so

a great risk of developing active TB if HIV-positive [28]. Households with a TB patient who is HIV-positive will tend to yield more active TB among screened contacts especially in lower-income countries [29,30]. This supports the idea of including HIV testing for contacts and the need to screen household contacts of HIV-positive TB patients as a priority, to detect TB early and identify contacts who can benefit from IPT. Recommendations by WHO on TB contact screening have emphasized this [9,29]. The most recent Malawian recommendations include IPT for all PLHIV, and so IPT could be provided as part of HIV care for all contacts diagnosed HIV positive [20,28,29].

The analysis of time to initiate of IPT found that patient-delivered screening resulted in more rapid TPT initiation among U5Y, such that over 50% of children had initiated TPT by day 40. Therefore, patient-delivered screening likely reduces mean time to initiation of IPT. This supports findings from other studies investigating patient-delivered screening for contact tracing in sexually-transmitted diseases [31,32].

Our subgroup analysis found that willingness and ability to participate in patient-delivered screening and IPT initiation was closely associated with household wealth (S1 Table). However, National programmes should recognize that poverty remains one of the key determinants of health in Malawi, and globally. It plays a major role in many TB control activities. TB is related to poverty with association to poor living conditions, poor nutrition status and poor health services [33–36]. We had anticipated that patient-delivered screening would be able to reach the poorest households but found that instead we introduced inequity. The reasons for this unanticipated finding need to be explored further. The intervention was highly utilized by those whose income was high which contrasted with the main aim of the intervention (S1 Table).

Limitations of this study includes the fact that we had a higher-than-expected refusal rate (51.7%) which limited study size and power of study. Thus, our participating households are not necessarily representative of all TB-affected households in Blantyre. We were not able to provide HIV testing and counselling to our household contacts, due to resource limitations. However, this would have provided estimates of HIV prevalence among household contacts and could also have increased TPT uptake in adults. Therefore, we cannot estimate HIV prevalence among TB contacts.

Potential contributions towards the lower-than-anticipated participation in this household level TB screening intervention include ill health of the index, and financial costs, and social stigma especially when a household visit is included. Lack of privacy within the Malawi National TB programme may exacerbate these concerns especially for patients who do have HIV-related TB. Screening approaches involving patient-delivered should be evaluated for effectiveness and acceptability in a range of setting with mixed methods approaches to further explore these barriers. Importantly, linked qualitative research in this study also underscores that acceptability was generally high, however, with many patients reporting strongly positive reflections on the value of each arm. In part we identified easily remedied logistical barriers such as the need to introduce contact screening while patients are waiting to be registered, and the need to offer additional support to patients who are too ill or too poor to cope with any additional facility visits.

In conclusion, we found that a patient-delivered household TB contact screening intervention resulted in greater and more rapid uptake of TB preventive therapy. For low-resource settings struggling to implement contact tracing interventions, patient-delivered tracing could be an effective approach to increasing access to TB care and prevention services.

## Supporting information

**S1 Checklist. Consort checklist.**
(DOCX)

**S1 Dataset. Data.**
(ZIP)

**S1 File. COMREC accepted protocol.**
(DOCX)

**S1 Table. IPT initiation among under year old household contacts.**
(DOCX)

## Acknowledgments

The authors would like to thank the participants; the leadership and authorities of Blantyre DHO, and QECH for permission to carry out the study. We also like to thank National TB Programme for allowing this study to take place. [†] Dr Ngwira passed away before the submission of this final manuscript. Dr Kaswaswa accept responsibility for the integrity and validity of the data collected and analysed.

## Author Contributions

**Conceptualization:** Kruger Kaswaswa, Geoffrey Chipungu, Elizabeth L. Corbett.

**Data curation:** Kruger Kaswaswa, Peter MacPherson, Mavuto Mukaka, Elizabeth L. Corbett.

**Formal analysis:** Kruger Kaswaswa, Peter MacPherson, Mavuto Mukaka, Elizabeth L. Corbett.

**Funding acquisition:** Jon Odland, Geoffrey Chipungu, Elizabeth L. Corbett.

**Investigation:** Kruger Kaswaswa, Moses Kumwenda, Deus Thindwa, Marriott Nliwasa, Mavuto Mukaka.

**Methodology:** Kruger Kaswaswa, Elizabeth L. Corbett.

**Project administration:** Kruger Kaswaswa, Mphatso Mwapasa, Tamiwe Tomoka, Geoffrey Chipungu.

**Resources:** Marriott Nliwasa, Mphatso Mwapasa, Jon Odland, Geoffrey Chipungu, Elizabeth L. Corbett.

**Software:** Peter MacPherson, Deus Thindwa, Marriott Nliwasa, Mavuto Mukaka.

**Supervision:** James Mpunga, Jon Odland, Tamiwe Tomoka, Mavuto Mukaka, Elizabeth L. Corbett.

**Validation:** Kruger Kaswaswa, Peter MacPherson, James Mpunga.

**Visualization:** Moses Kumwenda.

**Writing – original draft:** Kruger Kaswaswa, Peter MacPherson, Mavuto Mukaka, Elizabeth L. Corbett.

**Writing – review & editing:** Peter MacPherson, Moses Kumwenda, James Mpunga, Marriott Nliwasa, Jon Odland, Geoffrey Chipungu, Mavuto Mukaka, Elizabeth L. Corbett.

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
