## [Decision Letter · Decision Letter 0]

6 Sep 2021

PONE-D-20-11614Effect of patient-delivered household contact tracing and prevention for tuberculosis: a household cluster-randomised trial in MalawiPLOS ONE

Dear Dr. Kaswaswa,

Thank you for submitting your manuscript to PLOS ONE. After careful consideration, we feel that it has merit but does not fully meet PLOS ONE’s publication criteria as it currently stands. Therefore, we invite you to submit a revised version of the manuscript that addresses the points raised during the review process. Many thanks for submitting this manuscript to the journal. It's a really interesting trial and it's impressive how you implemented this research within the national TB program. Apologies for the length of time it's taken to get this decision to you. I was just recently asked to take on the academic editor role, it seems that the journal office had initially obtained the reviews before assigning an editor. There are three helpful reviews and I would encourage you to address the comments made by all three. I would just like to make two points where I particularly agree with the reviewers, and one additional point where it may just be me that got confused:I agree with the comment from reviewer 2 that the introduction could be substantially shortened to just capture the key background and rationale for the intervention and the studyReviewer 3 makes a good point that the time-to-event analyses appear in the Results section but not in the Methods section. If I understand correctly, these analyses were not predefined in the trial protocol or statistical analysis plan - therefore it would be helpful in the Methods section to just briefly explain that and the rationale for doing these post hocI found the two sentences in lines 59-63 of the abstract confusing. There seemed to be two similar key results being presented here. I may have missed something but I couldn't see exactly what the difference between the two sentences was, and I couldn't tie the second sentence and the adjusted odds ratio to anything that appears in the main manuscript or tables. I would suggest restricting yourself to only presenting the two primary outcome analyses in the abstract.Please submit your revised manuscript by Oct 21 2021 11:59PM. If you will need more time than this to complete your revisions, please reply to this message or contact the journal office at plosone@plos.org. Please include the following items when submitting your revised manuscript:A rebuttal letter that responds to each point raised by the academic editor and reviewer(s). You should upload this letter as a separate file labeled 'Response to Reviewers'.A marked-up copy of your manuscript that highlights changes made to the original version. You should upload this as a separate file labeled 'Revised Manuscript with Track Changes'.An unmarked version of your revised paper without tracked changes. You should upload this as a separate file labeled 'Manuscript'.If applicable, we recommend that you deposit your laboratory protocols in protocols.io to enhance the reproducibility of your results. Protocols.io assigns your protocol its own identifier (DOI) so that it can be cited independently in the future. For instructions see: https://journals.plos.org/plosone/s/submission-guidelines#loc-laboratory-protocols. Additionally, PLOS ONE offers an option for publishing peer-reviewed Lab Protocol articles, which describe protocols hosted on protocols.io. Read more information on sharing protocols at https://plos.org/protocols?utm_medium=editorial-email&utm_source=authorletters&utm_campaign=protocols.

We look forward to receiving your revised manuscript.

Kind regards,

Richard John Lessells, BSc, MBChB, MRCP, DTM&H, DipHIVMed, PhD

Academic Editor

PLOS ONE

Journal Requirements:

3. Thank you for submitting your clinical trial to PLOS ONE and for providing the name of the registry and the registration number. The information in the registry entry suggests that your trial was registered after patient recruitment began. PLOS ONE strongly encourages authors to register all trials before recruiting the first participant in a study.

a) your reasons for your delay in registering this study (after enrolment of participants started);

b) confirmation that all related trials are registered by stating: “The authors confirm that all ongoing and related trials for this drug/intervention are registered.

4. PLOS requires an ORCID iD for the corresponding author in Editorial Manager on papers submitted after December 6th, 2016. Please ensure that you have an ORCID iD and that it is validated in Editorial Manager. To do this, go to ‘Update my Information’ (in the upper left-hand corner of the main menu), and click on the Fetch/Validate link next to the ORCID field. This will take you to the ORCID site and allow you to create a new iD or authenticate a pre-existing iD in Editorial Manager. Please see the following video for instructions on linking an ORCID iD to your Editorial Manager account: https://www.youtube.com/watch?v=_xcclfuvtxQ.

5. We note that you have referenced "Malawi Government(GOM). July 2012-June 2013 Annual Tuberculosis Report (Unpublished Report).Lilongwe: National TB Programme,Malawi Government; 2013" and "Malawi Government(GOM). July 2013-June 2014 Annual Tuberculosis Report (Unpublished Report).Lilongwe: National TB Programme,Malawi Government; 2014.". which has currently not yet been accepted for publication. Please remove this from your References and amend this to state in the body of your manuscript: "Malawi Government(GOM). July 2012-June 2013 Annual Tuberculosis Report (Unpublished Report).Lilongwe: National TB Programme,Malawi Government; 2013" and "Malawi Government(GOM). July 2013-June 2014 Annual Tuberculosis Report (Unpublished Report).Lilongwe: National TB Programme,Malawi Government; 2014.". as detailed online in our guide for authors

6. Please include captions for your Supporting Information files at the end of your manuscript, and update any in-text citations to match accordingly. Please see our Supporting Information guidelines for more information: http://journals.plos.org/plosone/s/supporting-information

Reviewers' comments:

Reviewer's Responses to Questions

**Comments to the Author**

1. Is the manuscript technically sound, and do the data support the conclusions?

Reviewer #1: Yes

Reviewer #2: Yes

Reviewer #3: Partly

2. Has the statistical analysis been performed appropriately and rigorously? 

Reviewer #1: I Don't Know

Reviewer #2: Yes

Reviewer #3: Yes

3. Have the authors made all data underlying the findings in their manuscript fully available?

Reviewer #1: Yes

Reviewer #2: Yes

Reviewer #3: Yes

4. Is the manuscript presented in an intelligible fashion and written in standard English?

Reviewer #1: Yes

Reviewer #2: No

Reviewer #3: Yes

5. Review Comments to the Author

Reviewer #1: This is a very interesting study, which is well-planned and well written-up, from the TB programme and LSHTM researchers in Malawi. It has encouraging findings, though with limited uptake (which they correctly mention needs further attention prior to further implementation/ evaluation).

The study population and context issues are very well described, as are all other parts of the paper.

The patient delivered TB contact tracing intervention ('PACTS) was a well designed, feasible and low-cost intervention in Malawi context. Indeed it should also be also be feasible in other low income, high TB prevalence settings in sub Saharan Africa.

The control is usual care (asking patients to send their household contacts to be examined at the health facility, which is difficult due to distance and cost in this setting), and was clearly described.

The outcome results are clearly described and presented. However, in the Conclusion I suggest they add into the sentance 'An alternative patient-delivered symptom screen and IPT eligibility assessment significantly increased timely IPT uptake among U5Y children, [ADD] but not a significantly increased TB diagnosis. ....

The authors could be a bit more explicit in defining the (patients' household) clusters in the study design section, as I initially wondered whether this should really be descried as a cluster RCT. So I asked my medical statistician colleague to have a look, and when he read the initial methods he also at first wondered if it is appropriate to call this a cluster RCT. That is, we both found it a little confusing initially, but now agree that it is a cluster RCT because they randomised index cases plus their households, and the outcomes are based on data from all household members.

The ethics, of obtaining data from the household members, was addressed and approval addressed that and they did ultimately obtain consent from them too.

Please note that I reviewed the paper from the perspective of experience in TB control in LMICs/Africa, and experience in conducting trials. While the statistics looked appropriate to me, I am not a statistician and I did not check in these statistical details. My statistian colleague had a look at whether it is appropriately called a cluster RCT, but he did not do the rest of a statistical review. I assume that you will have a statistician review fully.

Minor points: There were not typos or other minor problems in the script. A very small point, the authors could check and change household to households in some places.

I commend the authors for designing, conducting and writing up this intervention/ trial so expertly.

I suggest, after a satisfactory statistical review, and after addressing my minor revisions, that it be accepted for publication.

Reviewer #2: 1. Line 49 Abstract seems to contain an error: “…. proportion of all adult of TB treatment…..”

2. Line 51. What is “U5Y”? (Presume under 5 year old children).

3. Lines 80-82: something seems amiss in this sentence: “In rural South Africa, prevalence of TB in household contacts was high compared to household contacts without TB (6,075 per 100,000 versus 407 per 100,000), with most contacts with culture-confirmed TB being asymptomatic.” Do you mean “compared to people living in households without a known TB case”?

4. In my view the introduction could be substantially reduced in length. Only three paragraphs are needed:

a. Importance of contact investigation for TB control in high burden settings

b. Limitations of existing model for implementing contact investigation

c. Rationale for proposed new model that is to be evaluated here and objective the present investigation.

5. Line 140: Study design. I presume the authors mean “open label” (that is, unblinded).

6. Line 144: TB prevalence cannot have been “> 900,000 per 100,000”.

7. Line 150: needs re-wording. Suggest “Adults with pulmonary TB who registered for treatment at either site …..”

8. Line 303-305 contains an error: “apart from for sputum microbiology where index cases in the SOC arm were more likely to have microbiologically-confirmed disease than those in the SOC arm,” (The second “SOC” should be “PACT”.)

9. Line 321-322: It looks like odds ratios have been calculated with the SOC (control) group as the risk group. Is that correct? More intuitive to treat the experimental group (PACT) as the risk group.

10. Lines 367ff: Not particularly helpful to re-state the results in the first paragraph of the discussion (BTW, the value of the aOR in line 368 (0.17) is incorrect). It would be helpful to simply state the answer the study questions / objectives outline in the last paragraph of the Introduction. It seems to me that the PACT intervention did not increase case finding in household contacts. It did increase initiation, but not completion, of IPT in young children. It also increased some testing for TB.

Reviewer #3: I thought this methods utilized in this manuscript were generally good, but with some confusing parts. I also got really confused about the time-to-event model that seemed to appear out of nowhere in the discussion. Also, some editing will be needed.

1. The language is a little sloppy at times throughout. I encourage you to have this evaluated by someone for language and grammer. If you have already, whoever did it is not good enough. A couple examples:

a. (line 59-60) "PACTS arm…were significantly more likely…" Verb agreement is incorrect and, frankly I believe the subject of the sentence does not make sense.

b. (line 151) "…screened for eligible…" is incorrect.

2. (lines 270-272) Please provide a methodological citation for the method used.

3. In addition, please cite the software utilized for these analyses (if any) as well as the level of significance.

4. (line 336) Instead of "not significantly likely to complete", I suggest saying, "not at significantly higher odds of completing".

5. (lines 349-350) This is too unstable to report, likely because of the zero count. This is tough in the context of a mixed model. Ideally one would run an exact method here, but as far as I know, exact methods will not allow you to include random effects or anything to account for the clustering. I'm guessing that's the way to go. The CI will be so wide with or without random effects that I would just run an exact method and note that it was run this way.

6. (lines 367-373) I was confused about which model is which here since all outcomes are individual-level. Maybe you can tie these into what is mentioned from the primary and secondary outcomes sections.

7. (line 370) There was no time-to-event analysis mentioned in the methods. If these were analyses run for this manuscript, please include them in the methods and probably a table is needed. If these are from a different manuscript, please cite that manuscript here.

6. PLOS authors have the option to publish the peer review history of their article (what does this mean?). If published, this will include your full peer review and any attached files.

Reviewer #1: **Yes: **John Walley, Professor of International Public Health, Nuffield Centre for International Health and Development, University of Leeds, UK.

Reviewer #2: **Yes: **Guy B Marks

Reviewer #3: No

---

## [Author Response · Author response to Decision Letter 0]

9 Dec 2021

We have addressed the issues raised by the reviewer in the communication letter under the "Responses to reviewer's" file

---

## [Decision Letter · Decision Letter 1]

2 Mar 2022

PONE-D-20-11614R1Effect of patient-delivered household contact tracing and prevention for tuberculosis: a household cluster-randomised trial in MalawiPLOS ONE

Dear Dr. Kaswaswa,

Thank you for submitting your manuscript to PLOS ONE. After careful consideration, we feel that it has merit but does not fully meet PLOS ONE’s publication criteria as it currently stands. Therefore, we invite you to submit a revised version of the manuscript that addresses the points raised during the review process. There are just a few remaining minor points for you to address from the reviewers. If you can please do your best to attend to these in a revision then hopefully we can move to a final acceptance.  Please submit your revised manuscript by Apr 16 2022 11:59PM. If you will need more time than this to complete your revisions, please reply to this message or contact the journal office at plosone@plos.org. Please include the following items when submitting your revised manuscript:A rebuttal letter that responds to each point raised by the academic editor and reviewer(s). You should upload this letter as a separate file labeled 'Response to Reviewers'.A marked-up copy of your manuscript that highlights changes made to the original version. You should upload this as a separate file labeled 'Revised Manuscript with Track Changes'.An unmarked version of your revised paper without tracked changes. You should upload this as a separate file labeled 'Manuscript'.If applicable, we recommend that you deposit your laboratory protocols in protocols.io to enhance the reproducibility of your results. Protocols.io assigns your protocol its own identifier (DOI) so that it can be cited independently in the future. For instructions see: https://journals.plos.org/plosone/s/submission-guidelines#loc-laboratory-protocols. Additionally, PLOS ONE offers an option for publishing peer-reviewed Lab Protocol articles, which describe protocols hosted on protocols.io. Read more information on sharing protocols at https://plos.org/protocols?utm_medium=editorial-email&utm_source=authorletters&utm_campaign=protocols.

We look forward to receiving your revised manuscript.

Kind regards,

Richard John Lessells, BSc, MBChB, MRCP, DTM&H, DipHIVMed, PhD

Academic Editor

PLOS ONE

Journal Requirements:

Reviewers' comments:

Reviewer's Responses to Questions

**Comments to the Author**

1. If the authors have adequately addressed your comments raised in a previous round of review and you feel that this manuscript is now acceptable for publication, you may indicate that here to bypass the “Comments to the Author” section, enter your conflict of interest statement in the “Confidential to Editor” section, and submit your "Accept" recommendation.

Reviewer #1: All comments have been addressed

Reviewer #2: (No Response)

Reviewer #3: All comments have been addressed

2. Is the manuscript technically sound, and do the data support the conclusions?

Reviewer #1: Yes

Reviewer #2: Yes

Reviewer #3: (No Response)

3. Has the statistical analysis been performed appropriately and rigorously? 

Reviewer #1: I Don't Know

Reviewer #2: Yes

Reviewer #3: (No Response)

4. Have the authors made all data underlying the findings in their manuscript fully available?

Reviewer #1: Yes

Reviewer #2: Yes

Reviewer #3: (No Response)

5. Is the manuscript presented in an intelligible fashion and written in standard English?

Reviewer #1: No

Reviewer #2: Yes

Reviewer #3: (No Response)

6. Review Comments to the Author

Reviewer #1: I have found it difficult to review the revisions made, as there is no file with track changes or other details of the revisions.

Also I'm constrained doing the review as I'm currently on extended leave overseas and working from a tablet not computer in a guest house.

I still feel this paper should be published, but with minor revisions, as I've noticed another problem. The secondary outcome descriptions, i and iv (from line 208) are not clearly diffierent Furthermore the presentation of these secondary outcomes in the results text and tables 2 and 3 don't seem to match in their order and description. Please review and edit for clarifty and consistency.

Reviewer #2: Most of the comments that I made previously have been adequately addressed. There are some matters still to be addressed.

1. Some improved clarity is still needed. Suggestion for the Abstract as follows. Line 42-44: Would be better written as “Between May and December, 2014, we randomised patients with TB who consented to participate in the trial to either standard of care (SOC) or intervention (PACTS) arms. Participants randomised to PACTS received ….”. Line 47: “Patients randomised to SOC were managed in accordance with national guidelines, that is, they received verbal instruction …..”. Line 50-52: “The primary outcome was the proportion of adult contacts receiving treatment for TB within 3 months of randomisation. Secondary outcomes were the proportions of child contacts under age 5 years (U5Y) who were commenced on, and completed, TPT. Data were analysed by logistic regression with random effects to adjust for household clustering.”.

2. Introduction is much improved.

3. It would be helpful to cite the number of people who met the eligibility criteria (464) in the results text (line 267).

4. I note the response in relation to using the intervention group (PACTS) as the reference group for odds ratios. I do not agree. In essence, the reference group should be the control group. A “risk” or odds ratio that is less than one is readily interpreted as indicating that the intervention is protective, which is what you are seeking to achieve. If you insist on using the intervention group as the reference for the expression of odds ratios, you should at least make this clear in the methods and in the results and tables when you present the results. Otherwise, most people will misinterpret them.

5. In table 2 you need a footnote to show what covariates adjusted for.

Reviewer #3: (No Response)

7. PLOS authors have the option to publish the peer review history of their article (what does this mean?). If published, this will include your full peer review and any attached files.

Reviewer #1: No

Reviewer #2: **Yes: **Guy B Marks

Reviewer #3: No

---

## [Author Response · Author response to Decision Letter 1]

19 Apr 2022

We have attached the reviewer's response as part of the documents. See attached files.

---

## [Editor Report · Decision Letter 2]

18 May 2022

Effect of patient-delivered household contact tracing and prevention for tuberculosis: a household cluster-randomised trial in Malawi

PONE-D-20-11614R2

Dear Dr. Kaswaswa,

We’re pleased to inform you that your manuscript has been judged scientifically suitable for publication and will be formally accepted for publication once it meets all outstanding technical requirements.

Kind regards,

Richard John Lessells, BSc, MBChB, MRCP, DTM&H, DipHIVMed, PhD

Academic Editor

PLOS ONE
---

## [Editor Report · Acceptance letter]

16 Aug 2022

PONE-D-20-11614R2 

Effect of patient-delivered household contact tracing and prevention for tuberculosis: a household cluster-randomised trial in Malawi 

Dear Dr. Kaswaswa:

I'm pleased to inform you that your manuscript has been deemed suitable for publication in PLOS ONE. Congratulations! Your manuscript is now with our production department. 

Kind regards, 

on behalf of

Dr. Richard John Lessells 

Academic Editor

PLOS ONE